# Enhanced Adipogenic Differentiation of Human Dental Pulp Stem Cells in Enzymatically Decellularized Adipose Tissue Solid Foams

**DOI:** 10.3390/biology11081099

**Published:** 2022-07-23

**Authors:** Nerea Garcia-Urkia, Jon Luzuriaga, Veronica Uribe-Etxebarria, Igor Irastorza, Francisco Javier Fernandez-San-Argimiro, Beatriz Olalde, Nerea Briz, Fernando Unda, Gaskon Ibarretxe, Iratxe Madarieta, Jose Ramon Pineda

**Affiliations:** 1TECNALIA, Basque Research and Technology Alliance (BRTA), 20009 Donostia-San Sebastian, Spain; nerea.garcia@tecnalia.com (N.G.-U.); xabier.fernandez@tecnalia.com (F.J.F.-S.-A.); beatriz.olalde@tecnalia.com (B.O.); nerea.briz@tecnalia.com (N.B.); 2Cell Biology and Histology Department, University of the Basque Country (UPV/EHU), 48940 Leioa, Spain; jon.luzuriaga@ehu.eus (J.L.); vero18791@gmail.com (V.U.-E.); igor.irastorza@ehu.eus (I.I.); fernando.unda@ehu.eus (F.U.); gaskon.ibarretxe@ehu.eus (G.I.); 3Achucarro Basque Center for Neuroscience, 48940 Leioa, Spain

**Keywords:** human adipose tissue, decellularization, extracellular matrix, human dental pulp stem cells, solid foam, adipogenic differentiation, stem cell culture, personalized medicine

## Abstract

**Simple Summary:**

Obesity shortens human lifespan and represents one of the most important public health problems causing significant economic and societal consequences worldwide. However, the current development of physiological human 3D adipose tissue models for in vitro research on preclinical personalized medicine is limited and expensive. Here, we designed, produced, and characterized 3D solid foams using a mixture of bovine collagen I and decellularized human adipose tissue to serve as a 3D matrix mimicking in vivo adipose microenvironment for cell culture purposes. Furthermore, we sought to validate its compatibility for the culture of human mesenchymal stem cells isolated from the dental pulp. We demonstrated that 3D solid foams are able to integrate the stem cells from the dental pulp and provide the appropriate cues to differentiate them into mature adipocytes. The results represent an advance of in vitro 3D models using a human extracellular matrix derived material for future personalized stem cell therapies.

**Abstract:**

Engineered 3D human adipose tissue models and the development of physiological human 3D in vitro models to test new therapeutic compounds and advance in the study of pathophysiological mechanisms of disease is still technically challenging and expensive. To reduce costs and develop new technologies to study human adipogenesis and stem cell differentiation in a controlled in vitro system, here we report the design, characterization, and validation of extracellular matrix (ECM)-based materials of decellularized human adipose tissue (hDAT) or bovine collagen-I (bCOL-I) for 3D adipogenic stem cell culture. We aimed at recapitulating the dynamics, composition, and structure of the native ECM to optimize the adipogenic differentiation of human mesenchymal stem cells. hDAT was obtained by a two-enzymatic step decellularization protocol and post-processed by freeze-drying to produce 3D solid foams. These solid foams were employed either as pure hDAT, or combined with bCOL-I in a 3:1 proportion, to recreate a microenvironment compatible with stem cell survival and differentiation. We sought to investigate the effect of the adipogenic inductive extracellular 3D-microenvironment on human multipotent dental pulp stem cells (hDPSCs). We found that solid foams supported hDPSC viability and proliferation. Incubation of hDPSCs with adipogenic medium in hDAT-based solid foams increased the expression of mature adipocyte LPL and c/EBP gene markers as determined by RT-qPCR, with respect to bCOL-I solid foams. Moreover, hDPSC capability to differentiate towards adipocytes was assessed by PPAR-γ immunostaining and Oil-red lipid droplet staining. We found out that both hDAT and mixed 3:1 hDAT-COL-I solid foams could support adipogenesis in 3D-hDPSC stem cell cultures significantly more efficiently than solid foams of bCOL-I, opening the possibility to obtain hDAT-based solid foams with customized properties. The combination of human-derived ECM biomaterials with synthetic proteins can, thus, be envisaged to reduce fabrication costs, thus facilitating the widespread use of autologous stem cells and biomaterials for personalized medicine.

## 1. Introduction

Changes in the lifestyle and dietary habits over the last decades have resulted in a burst of a worldwide obesity pandemic that shortens the lifespan of the affected individuals. Obesity and related diseases, including cardiovascular diseases, diabetes, musculoskeletal disorders, fatty liver, and even some type of cancers [1,2], currently represent one of the most important public health problems with significant economic and societal consequences world-wide [3]. Obesity and related diseases produce 2.8 million deaths annually worldwide with an estimated medical cost of 2 trillion dollars [4,5]. Most of the available non-invasive medical therapies for obesity are non-efficient in a long-term evaluation, and one pharmacological prescription that works well in a subset of patients might be useless with another; therefore, there is a constant need for new personalized therapeutic strategies that take into account individual genetic variability [3,6].

One of the most promising strategies to advance in personalized anti-obesity treatments is the development of physiological human three-dimensional (3D) tissue culture models for drug testing, regarding the regulation of adipogenesis and adipose cell differentiation [7,8,9]. Traditional two-dimensional (2D) cell cultures have served biologists well for decades, but this type of culture does not represent the physiological reality. Cells cultured in 2D are more susceptible to the drug effects and show differences in surface receptor organization which, among others, produce a decrease in drug efficiency and contributes to drug resistance [10]. In contrast, it is widely demonstrated that 3D cell cultures reproduce more accurately the complexity of native tissues and improve cell morphology, proliferation, and differentiation. Recent investigations demonstrated that 3D tissue models support discoveries into the mechanisms of adipose-related diseases and the development of novel anti-obesogenic therapeutic compounds [9,11,12,13].

There is a variety of methodologies to produce in vitro 3D tissue models, which, typically, required the use of biomaterials as scaffolds for cells to grow. Scaffold-based 3D cell cultures are expected to yield results with higher predictive value for clinical outcomes, and are well suited to drug discovery to obtain more accurate results [14,15]. Although substantial advances have been made, the development of an ideal tissue engineered human 3D adipose tissue model is still technically challenging.

The extracellular matrix (ECM) represents the secreted product of the resident cells of each tissue and organ, including both bioactive and structural molecules arranged in a unique 3D ultrastructure that supports the phenotype and the function of cells [16,17]. Cells and their surrounding ECM are in a continuous “dynamic reciprocity”, with cells responding to signals in the ECM to alter their behavior, and the cells, in turn, modifying the organization and composition of the ECM [18,19]. As a natural template for cells, ECM-derived materials are gaining clinical importance and market space due to their biochemical superiority and the capacity to promote functional tissue regeneration [20,21]. There are numerous commercially available biologic scaffold materials derived from a variety of tissues which have been successfully applied in both preclinical and clinical studies [22]. The goal of a decellularization protocol is to efficiently remove the cellular and nuclear material while minimizing any adverse effect on the composition, biological activity, and mechanical integrity, by retaining the 3D ultrastructure and composition of the native ECM [20,23]. A wide range of methods for decellularizing almost all types of tissues have been published [24,25], which typically involve a combination of physical, chemical, and enzymatic treatment of tissues. However, each treatment affects the biochemical composition, ultrastructure, and mechanical behavior of the remaining ECM materials, which, in turn, brings functional and pathophysiological implications to its regenerative capacities [21]. So, to remove as much cellular material as possible with minimal adverse effects on the resulting ECM material, a fine-tuned decellularization protocol tailored to the tissue of interest would be necessary [23]. To circumvent limitations of applying decellularized tissues in their intact form, such as limited tuned shape, porosity, and stiffness, as well as their mode of delivery for in vivo applications, numerous research groups are applying innovative processing methods to generate custom-made scaffold formats. Decellularized tissues can also be subjected to enzymatic digestion to fabricate ECM-derived hydrogels, foams, microcarriers, and coatings [23,24,25], as well as to synthesize bioinks for 3D bioprinting [26,27].

Our group patented a two-step enzymatic protocol for AT decellularization [28] and investigated the macrophage polarization and immunocompetent capacity of decellularized adipose tissue (DAT) obtained by enzymatic-based decellularization protocols. Our results evidenced the effective decellularization and conservation of ECM proteins as well as an absence of macrophage pro-inflammatory response, when these materials were processed as coatings [26,29].

Due to their unique capabilities to self-renew (stemness) and to differentiate toward one or more specialized cell types (potency), stem cells are being employed for reconstructing tissue/organ structure and function in vitro, representing a versatile source of cellular substitutes for a wide range of applications for both basic research and regenerative medicine [30,31,32,33,34]. Specifically, human dental pulp stem cells (hDPSC) are multipotent stem cells which can be differentiated to various linages, such as adipocytes, osteoblasts, and vascular and neural cells [35,36,37,38]. hDPSC have a high proliferation capacity and they are easily isolated, manipulated, expanded in vitro, and stored, making them a precious tool for tissue engineering and regenerative therapies [39], and the production of human-derived 3D tissue models. Our group has recently in vitro combined hDPSCs with porcine and human DAT produced by organic solvent decellularization and processed as solid foams. Interestingly, human-derived DAT solid foams provided a different microenvironment for hDPSCs because 3D cultures in this material were more refractory to osteogenesis and had a better adipogenic capacity than porcine DAT [29]. One of the main advantages of hDAT solid foams is that white AT is also a relatively abundant and available source of human-derived raw-tissue material, which can be obtained from liposuction surgery. Because both hDAT and hDPSCs can be isolated from human donors under ATMP standards, it could be interesting to combine them for personalized medicine applications.

In the present study, we followed a different two-enzymatic step decellularization protocol to obtain hDAT [26,28], which was processed by freeze-drying as solid foams, and combined with hDPSCs to investigate the inductive capability of these biologic scaffolds on the adipogenesis of hDPSCs, recreating a natural biochemical environment. The compatibility and interactions between adipogenic hDPSCs and the hDAT solid foams were assessed by comparing them to commercial single-protein-based scaffolds of bovine type I collagen (bCOL-I), which have already been extensively used for 3D modeling of tissues in vitro [29,40].

## 2. Materials and Methods

### 2.1. Human Adipose Tissue Decellularization

Following authorization AC20131754 from the French Ministry of Higher Education and Research, human AT was obtained from the biotech company Biopredict International (Saint Grégoire, France). The ATwas cleaned, creamed, and stored at −20 °C. AT was decellularized following the enzymatic digestion methodology previously published by our group [26]. Briefly, tissue was dispersed and homogenized in ultrapure water using aPolytron PT3100 device at 12,000 rpm for 5 min, and centrifuged at 900× *g* for 5 min to discard the liquid phase containing the lipids and keeping the protein pellets. Next, pellets were processed by a two-step enzymatic protocol previously described by the group [28] using lipoprotein lipase (Merck Life Science SL, Madrid, Spain) and Benzonase^®^ (Emprove^®^-bio, Merck Life Science SL, Madrid, Spain). Samples were cleaned with Phosphate Buffer Saline (PBS, Merck Life Science SL), supplemented with 1% (*v*/*v*) of Penycyline/Streptomycine and Fungizone (Gibco-BRL, Paisley, UK) and washed with Ultrapure milli Q water. Then, using a mixer mill (Retsch MM400, Haan, Germany) and a vacuum dessicator, samples were dried to obtain a fine-grained powder that was conserved at 4 °C until use.

### 2.2. Characterization of hDAT

hDAT was analyzed to assess the presence of residual DNA and lipids as well as the protein composition, as previously described [26]. DNA was extracted using a QiAmp kit (Qiagen, Madrid, Spain) following the manufacturer’s instructions, and absolute quantification was performed by quantitative reverse-transcription and real-time polymerase chain reaction (qRT-PCR) based on standard curves of known human DNA samples.

The total lipid content was determined using liquid chromatography coupled with Q-TOF mass spectrometry (UPLC/Q-TOF MS). Samples were injected into a Waters column (Acquity UPLC HSS T3 1.8 μm, 100 mm × 2.1 mm) and then heated to 65 °C to give rise to two mobile phases. The phase A and B consisted of a proportion of 40:60 *v*/*v* and 10:90 *v*/*v* respectively of acetonitrile and water with 10 mM ammonium acetate. The injection volume was 5 μL at 0.5 mL/min flow rate. UHPLC-MSE data were acquired on a SYNAPT G2 HDMS equipped with an electrospray ionization (ESI) source using a quadrupole time of flight (Q-ToF) setup.

Finally, the protein composition was analyzed using a liquid chromatography coupled with a tandem mass spectrometry (LC-MS/MS) in native ATs and hDAT. Samples were processed by homogenization in 8 M of urea, using a Precellys^®^ 24 homogenizer (Bertin Technologies, Brussels, Belgium). Thereafter, they were sonicated for 3 min, and centrifuged at 16,000× *g* for 10 min. After the clarification proteins were reduced, alkylated (with 5 mM DTT and 15 mM iodoacetamide, respectively) and digested with trypsin (0.01 μg/μL). Finally, the isolated peptides were desalted using C-18 Micro SpinColumns (Harvard J. Fungi 2021, 7, 392 4 of 18 Apparatus, Holliston, MA, USA) and analyzed by LC-MS/MS with a Q Exactive instrument (Thermo Fisher Scientific, Madrid, Spain).

### 2.3. hDAT Processing as Solid Foam

hDAT and bCOL-I (Ref. C9879; Merck KGaA, Darmstadt, Germany) solid foams were obtained by the freeze-drying method and following the formulations described in Table 1. 0.5 % (*w*/*v*) milled hDAT and bCOL-I were added to a 0.5 mol/L acetic acid solution and homogenized by magnetic stirring for 48 h at room temperature. After that, different moulds were used to prepare solid foams. For SEM, mechanical, and degradation testing, 12-well plates (Ref. CLS3513-50EA; Corning Incorporated, New York, NY, USA) with 1 mL were used, for swelling and qRT-PCR assays, 48-well polystyrene tissue culture treated plates (Ref. 056204; Nalge Nunc International, Rochester, NY, USA) were used with 225 µL solutions. For viability and proliferation assays Millicell EZ-slide 8-well glass slides (PEZGS0816; Merck Millipore, Burlingtone, MA, USA) and µ-slide angiogenesis devices (Ref. 81506; Ibidi GmbH, Gewerbehof Gräfelfing, Germany) were used respective with approximately 200 µL/cm^2^. To obtain the solid foams, all the samples were frozen at −20 °C overnight and freeze-dried at −10 °C and 63 Pa of pressure during 72 h. All the material was sterilized using ethylene oxide (Esterilizacion Sl, Barcelona, Spain) for 270 min at 38 °C and 40% relative humidity and kept at 4 °C in a vacuum desiccator until use in cell culture.

### 2.4. Scanning Electron Microscopy (SEM)

Solid foams microarchitecture and porosity were assessed by SEM. Samples were freeze-dried, pulse-coated using a JFC-1100 ion sputter (JEOL, Tokyo, Japan) with gold 85–10 nm, mounted, and then visualized with an accelerating voltage of 10 kV using a JEOL JSM-5910 LV SEM (JEOL Ltd., Tokyo, Japan).

### 2.5. Swelling Properties

Solid foam swelling properties were determined by a water absorption assay (n = 3). The dry weight (Wd) was recorded from lyophilized foams prior to bringing them to maximum hydration in distillated water for 24 h. Thereafter, the foams were carefully blotted to remove the excess liquid and the wet weight (Ws) was recorded. The equilibrium water content (%ECW) and mass swelling ratio (S) were determined with the following formulas:$$\%\mathrm{ECW}=\frac{\mathrm{Ws}-\mathrm{Wd}}{\mathrm{Ws}}\times 100\;\;\;\;S=\frac{\mathrm{Ws}-\mathrm{Wd}}{\mathrm{Wd}}$$

### 2.6. Mechanical Test

Mechanical properties of the hydrated foams were measured by oscillatory shear rheology (n = 2). Rheological experiments were carried out using a parallel-plate geometry (200 mm diameter steel with a gap of 1 mm) of a Discovery HR20 rheometer (TA Instruments, New Castle, Denver, CO, USA). Firstly, stress amplitude sweeps were performed at a constant frequency of 0.1 Hz to fix the amplitude parameter for each sample and to ensure that subsequent data were collected in the linear viscoelastic regime. All measurements were taken at room temperature in constant deformation control mode over a frequency range from 0.01 to 10 Hz.

### 2.7. Degradation Test

The in vitro degradation of the solid foams was evaluated by weight measures (n = 5) before and after the incubation in PBS (Gibco™, Waltham, MA, USA) containing 0.01% collagenase (Gibco, BRL, UK) for 4 h at 37 °C. After the incubation, the samples were removed, rinsed with distilled water, dried under vacuum, and weighed. The degradation (%) was calculated using the following equation:$$\mathrm{Degradation}\;\left(\%\right)=\frac{Wo-Wt}{Wo}\times 100$$
where *Wo* is the initial foam weight and *Wt* is the foam weight after the incubation time (*t*). 

### 2.8. hDPSC In Vitro Culture

Primary cultures of hDPSCs were isolated from human third molars obtained from healthy donors between 18–40 years after written informed consent in compliance with the 14/2007 Spanish directive for Biomedical research, and the protocol approved by the CEISH committee (CEISH/M10/2020/172) of the University of the Basque Country (UPV/EHU). hDPSC isolation and culture were carried out as previously reported [41,42]. Briefly, hDPSCs were cultured in basal medium composed of Dulbecco’s modified Eagle’s medium (DMEM, Lonza 12-733; Basel, Switzerland) supplemented with 10% fetal bovine serum (Ref. SV30160.03; Hyclone, Ge Healthcare Life Sciences, Logan, UT, USA), 2 mM L-glutamine (G7513, Sigma, St. Louis, MO, USA), 100 U/mL penicillin and 150 µg/mL streptomycin antibiotics (15140-122, Gibco). Then, 1 × 10^6^ cells/mL (between passage 1 and 3) were transferred to cell culture plates with incorporated solid foams for 3D culture. Foams were firstly wetted on basal media for at least 30 min (37 °C, 5% CO_2_). Afterwards, 1 × 10^5^ hDPSCs were added dropwise and cell-foam constructs were cultured for 72 h in basal medium.

### 2.9. hDPSC Viability and Proliferation

Propidium Iodide (Sigma-Aldrich, Burlington, MA, USA) and Calcein-AM (Invitrogen, Waltham, MA, USA) and were used to assess cell viability after 72 h of culture. Cell containing constructs were washed with 1× PBS (Gibco, Waltham, MA, USA) and stained with Propidium iodide staining (1 mg/mL; Sigma-Aldrich, Burlington, MA, USA) and Calcein AM (5 μM in PBS 1×; Invitrogen, Waltham, MA, USA) for 30 min at 37 °C and 5% CO_2_ for 5 min at 37 °C and 5% CO_2_. Cell proliferation was assessed by immunofluorescence staining incubating the primary antibody Ki67 (1:300, Ref. ab15580, Abcam, Cambridge, UK) overnight. After three rinses in PBS, secondary antibody Alexa Fluor 555 (1:500, Ref. A32723, Invitrogen, Waltham, MA, USA) was incubated for 2 h and cell nuclei were stained with 4′,6-diamino-2-fenilindol (DAPI; 1:1000, Ref.D1306, Invitrogen, Thermo Fisher Scientific, Waltham, MA, USA). All samples were visualized by fluorescent microscopy (Olympus LSM800, Tokyo, Japan) and images were analyzed by ImageJ program [43]. Viability and proliferation percentage results were presented as the mean ± standard deviation of 10 random fields in three independent experiments calculated using the following equations:Viability (%)=(live cells/total cells)×100Proliferation (%)=(Ki67 positive nuclei/total nuclei)×100

### 2.10. Adipogenic Differentiation of hDPSC

To induce adipogenic differentiation 5 × 10^4^ hDPSCs were cultured for two days in basal medium, then separated in two parallel cohorts, one maintained in basal medium and the other grown in adipogenic medium for additional 14 days. The adipogenic differentiation medium consisted of regular culture medium supplemented with 1 µM dexamethasone (Ref. 265005; Merck KGaA, Darmstadt, Germany), 0.5 mM 3-Isobutyl-1-methylxanthine (IBMX) (Ref. I5879; Merck KGaA, Darmstadt, Germany) and 1 µg/mL insulin (Ref. 91077C; SAFC Biosciences).

### 2.11. RNA Extraction and qRT-PCR

After 14 days of culture in basal or adipogenic medium, cellular RNA was extracted for the solid foams (n = 3). Firstly, cell-laden foams were cleaned with 1× PBS (Gibco, Waltham, MA, USA) and RNA extraction was done using RNAqueous kit (Refs. AM1906/AM1931; Ambion Life technologies, Waltham, MA, USA) and purity was checked by the Nanodrop Synergy HT (BioTek Agilent, Santa Clara, CA, USA). cDNA (50 ng/µL) was obtained by reverse transcription of total extracted RNA using the iScript cDNA kit (Ref. 1708890; BioRad, Hercules, CA, USA). Primer pairs used were obtained through the PrimerBlast method (Primer Bank) and they are listed in Table 2.

Quantitative PCR was setup mixing 4.5 µL of SsoAdvanced Universal SYBR^®^ Green Supermix (Ref. 1725271; BioRad, Hercules, CA, USA) with 0.5 µL of primers (0.3125 µM) and 0.3 µL of cDNA (1.5 ng/µL) adjusting to a final volume reaction of 10 µL per well with Nuclease Free water. All the primers were checked for optimal efficiency (>90%). We checked the melting curve method of qPCR reactions to verify they yielded only one amplification product. *β-ACTIN* and *GAPDH* were used as internal controls and the relative expression of each gene was calculated using the standard 2^−ΔΔCt^ method [44]. All reactions were performed in triplicate, and qPCR was run on an ABI PRISM^®^ 7000 (Thermo Fisher Scientific, Waltham, MA, USA). Data were analyzed using CFX Manager™ software (BioRad, Hercules, CA, USA).

### 2.12. PPAR-Gamma Immunofluorescence and Quantification

Following 14 days of culture, solid foams were washed with 1× PBS and fixed with 4 % paraformaldehyde (PFA, Ref. 158127; Sigma, St. Louis, MO, USA) in PBS for 10 min at RT. After that, samples were incubated with 10 % goat serum (50197Z, Invitrogen, Carlsbad, CA, USA) for 10 min at RT. Then, the following primary antibody was applied at a dilution of 1:200 overnight at 4 °C in a solution of 0.1% Triton X-100 and 1% BSA in PBS: anti-PPARɣ (1:250 dilution, Abcam ab59256, Cambridge, UK). Thereafter, secondary antibody goat anti-rabbit Alexa Fluor 488 (1:500, A32731, Thermo Fisher Scientific, Waltham, MA, USA) was incubated for 2 h at room temperature and cell nuclei were counterstained with DAPI (1:1000, Ref. D1306, Invitrogen, Thermo Fisher Scientific, Waltham, MA, USA). Fluorescence images were taken by fluorescence Zeiss LSM800 microscope (Zeiss, Overkochen, Germany) coupled to a Nikon DS-Qi1 camera (Nikon, Tokyo, Japan). For PPAR-gamma quantification, an average of n = 17–18 cells were analyzed for each condition randomly selected from aleatory regions. Briefly, each cell was delimited generating individual R.O.I.s (regions of interest) and the labeling intensity for each cell was determined using the mean value measurement on ImageJ software.

### 2.13. Oil Red Staining and Quantification

Cell laden solid foams cultured for 14 days were stained at room temperature for 10 min with a solution of 0.5 % of 1-(2,5-dimethyl-4-(2,5-dimethylphenyl) phenyldiazenyl) azonapthalen-2-ol (Oil red) (Ref.O1516; Merck KGaA, Darmstadt, Germany) solution diluted in polyethylene glycol (PEG) 100% (Ref.151957; MP Biomedicals), as previously described [29,45]. Exceeding Oil red was washed with a 60% ispropanol (ref. I9516, Sigma-Aldrich, Burlington, MA, USA) and twice with distilled water for one minute. Cell nuclei were counterstained with DAPI (1:1000, Invitrogen, CA, USA) in PBS (Gibco™, Waltham, MA, USA). Images were acquired using an epifluorescence and transmission light Olympus IX71 microscope, coupled to an Olympus DP71 digital camera. For lipid droplet quantification, an average of n = 17–18 cells were analyzed for each condition randomly selected from aleatory regions. For each cell (region of interest or R.O.I.) droplets were identified using an inversion of the RGB color space from the color threshold, then automatic quantification was made using the “Analyze Particles” function from ImageJ v.1.53k software (Wayne Rasband and contributors, National Institutes of Health, Bethesda, MD 20814, USA).

### 2.14. Statistical Analysis

All data sets were subjected to normality test (Shapiro–Wilk test) and homoscedasticity (Levene test) to verify whether they fitted a normal distribution or not. Statistical comparisons between samples groups of three independent experiments were made compared to each respective basal media condition: one-way ANOVA and post-hoc Dunns or student’s *t*-test. Confidence intervals were set at 95% (*p* < 0.05) for all tests.

## 3. Results

### 3.1. hDAT and 3hDAT:1bCOL-I Solid Foams Degradation Is Slower Than bCOL-I

Based on our previous experience, in the present study human AT was decellularized by a two-step enzymatic method [26,28]. The resultant hDAT met the decellularization criteria (remnant DNA ≤ 50ng/mg and absence of cell nuclei) established by the research community [24,46] as well as by a recently published standard guide ASTM International (ASTM F3354-19) [28] based upon the findings of studies in which adverse cell and host responses have been avoided. The decellularization protocol described in this paper resulted in a hDAT that contained 2.38 ± 0.16 ng/mg dry weight remnant DNA without any presence of cell nuclei after DAPI staining. Besides, the protocol of tissue decellularization was very efficient for lipid removal, showing 0.8 ± 0.1 µg/mg of remnant lipids which corresponded to a 0.1% (*w*/*w*) with respect to the hDAM dry weight. Proteomic analysis of hDAT showed the conservation of 45 of 52 ECM specific proteins identified in native AT (86.5%), including all of the identified basement membrane proteins.

Once the decellularization criteria and composition were verified, hDAT-based solid foams were obtained following the formulations described in Table 1 to be used as scaffolds for in vitro 3D culture. We observed that hDAT and bCOL-I were optimally processed by the freeze-drying method to obtain solid foams, and SEM analysis revealed a highly interconnected porous mesh-like structures in all cases, with a pore size of 50–100 µm (Figure 1b). All the solid foams swelled properly without losing their geometry, and hDAT and 3hDAT:1bCOL-I solid foams had a slightly higher tendency to water absorption of 92.4 ± 0.9%, 92.1 ± 0.7%, respectively, in comparison to 90.3 ± 0.1% of bCOL-I solid foams, as shown in EWC and S results (Table 1).

The viscoelastic behaviour of the solid foams showed higher G′ (storage modulus) values than G″ (loss modulus) during the entire range of frequencies, indicating the predominance of the elastic behavior over the viscous one for all assayed samples. Solid foams showed higher average storage modulus with the increase in bCOL-I quantity in the formulation: 152.8 ± 15.6 Pa, 128.0 ± 11.1 Pa, and 302.9 ± 22.2 Pa, for hDAT, 3hDAT:1bCOL-I, and bCOL-I, respectively (Table 1). In the same way, the degradation rate increased accordingly with bCOL-I content in the solid foam formulation, showing a degradation of 22.9 ± 11.9%, 26.1 ± 11.5%, and 61.95 ± 15.99% for hDAT, 3hDAT:1bCOL-I, and bCOL-I after 4 h of collagenase treatment, respectively (Figure 1 and Table 1). Thus, water absorption results could explain the water contribution to the overall resistance of the materials to deformation, and are in agreement with reologícal test results that show greater elastic or storage modulus (G′) in bCOL-I solid foams and greater viscous or loss modulus (G″) in hDAT and 3hDAT:1bCOL-I solid foams.

### 3.2. hDPSC Viability and Proliferation

To assess cell viability in the different compositions of the hDAT, 3hDAT:1bCOL-I, and bCOL-I solid foams, freshly isolated and expanded hDPSCs were dissociated and cultured in the foams and were maintained for 72 h until labeling to allow cells to integrate into the foam. Next, a colabeling of Calcein and Propidium Iodide (PI) was made to detect both cell viability (cytoplamic staining) and cell death (PI incorporation). Our results showed similar values with 97.5 ± 3.5%, 99.4 ± 0.6%, and 95.9 ± 2.8% cell viability values for hDAT, 3hDAT:1bCOL-I, and bCOL-I samples, respectively (one-way ANOVA *p* = 0.4615. Figure 2a,b).

With regard to the ratio of cell proliferation analyzed by Ki67 staining, our results showed a hDPSC proliferation ratio of 12.1 ± 4.7%, 8.3 ± 2.6%, and 7.8 ± 4.0% for hDAT, 3hDAT:1bCOL-I, and bCOL-I solid foams, respectively, after 72h of culture (one-way ANOVA *p* = 0.3887. Figure 3a,b).

### 3.3. hDPSCs Cultured on Solid Foams Expressed Adipogenic Markers

To assess the change of gene expression pattern towards adipocyte phenotype, cell-laden solid foams were subjected to adipogenic induction/differentiation using adipogenic culture medium for 14 days. Adipogenic markers, such as Lipoprotein lipase (LPL), an enzyme necessary for intracellular lipid accumulation [47], and the CCAAT/enhancer binding protein (C/EBP) involved in adipocyte differentiation [48] were analyzed by RT-qPCR. We did not find any statistical difference for LPL in basal conditions; ratios of 1.000 ± 0.006, 0.327 ± 0.083, and 1.470 ± 0.532 for bCOL-I, 3hDAT:1bCOL-I, and hDAT, respectively (Figure 4a). However, when cultured with adipogenic medium, we found an increase of 7.573 ± 4.737 and 8.373 ± 1-fold change ratios for 3hDAT:1bCOL-I and hDAT, respectively, with statistically significant differences between each type of solid foam (Dunn’s post-hoc, one-way ANOVA * *p* < 0.05. Figure 4b). For C/EBP, in basal medium conditions we found fold change ratios of 0.918 ± 0.066, 2.636 ± 0.272, and 5.930 ± 2.461 for bCOL-I, 3hDAT:1bCOL-I, and hDAT, respectively (Figure 4c). When cultured in adipogenic medium conditions, C/EBP fold change values increased to 20.280 ± 4.790 and 13.317 ± 1.314 for 3hDAT:1bCOL-I and hDAT solid foams, respectively, with statistically significant differences (Dunn’s post-hoc, one-way ANOVA * *p* < 0.05. Figure 4c,d).

### 3.4. PPAR-γ Immunofluorescence and Oil Red Staining

The over-expression of genes involved in adipogenesis prompted us to check the presence of the peroxisome proliferator-activated receptor gamma (PPAR-γ), a receptor found in adipose tissue and involved in a terminal adipocyte differentiation [49]. Positive cells for immunofluorescence against PPAR-γ were detected in hDAT and 3hDAT:1bCOL-I solid foams with both basal and adipogenic medium (Figure 5a). Quantification of the optical signal intensity (o.d.) showed no nuclear staining and no significant differences for bCOL-I foams between basal and adipogenic medium (1005.8 ± 581.9 o.d. for basal and 3229.2 ± 2617.1 o.d. for adipogenic medium, student’s *t*-test *p* = 0.1344). However, 3hDAT:1bCOL-I foams showed an increase of immunolabeling from 3193.5 ± 1804.1 to 11,280.5 ± 2049.4 o.d. (student’s *t*-test *p* < 0.0005) and hDAT foams also increased from 4429.5 ± 1091.3 to 9865.9 ± 877. 4 o.d. with a more nuclear immunolabeling pattern (Figure 5a,b, student’s *t*-test *p* < 0.0001).

To fully corroborate the terminal adipocyte differentiation phenotype of hDPSCs seeded on the solid foams, the presence of neutral triglycerides and lipid droplets was detected using a type of Sudan dye known as Oil red. The Oil red staining was performed in parallel cellular cohorts cultured either in basal medium or adipogenic medium showing the pattern of lipid droplet accumulation around the nucleus of differentiated hDPSCs, consistent with their mature adipocyte phenotype (Figure 6a). Quantification of the number of droplets per region of interest (R.O.I.) showed values of 73.2 ± 14,67 o.d. for basal and 135.8 ± 63.25 o.d. for adipogenic medium for bCOL-I solid foams (student’s *t*-test *p* = 0.0866). 3hDAT:1bCOL-I foams showed an increase of droplet accumulation from 52.33 ± 22.82 to 153.8 ± 29.04 o.d. when cultured using adipogenic medium (student’s *t*-test *p* = 0.0179). In hDAT foams lipid droplet presence also increased from 73.5 ± 15.9 to 113 ± 24.14 particles (Figure 6a,b, student’s *t*-test *p* = 0.026).

## 4. Discussion

The worldwide prevalence of obesity has nearly tripled since 1975, and, currently, 60% of citizens in Europe are either overweight or obese [50]. AT dysfunctions contribute to obesity-related metabolic diseases. Indeed, obesity has been reported to be linked to a decline in the mean life expectancy due to its association with comorbidities, such as atherosclerosis, or increasing the risk of developing several pathologies, such as type 2 diabetes, hypertension, coronary heart disease, stroke, fatty liver disease, inflammation, and neurocognitive functions, among others [51,52,53,54]. Moreover, COVID-19 has been associated with worse outcomes in obese people [55]. For all these reasons, there is an urgent need for the development of in vitro humanized AT models. Furthermore, these models should reproduce aspects of the native tissue to optimize and speed up research in its pathophysiology and personalized drug-screening, offering major benefits and advances in the study of pathophysiological mechanisms of disease to the biomedical research community and Health Systems.

Native tissue ECM is the natural cell-template to be recreated for scaffolding in 3D culture models. Thanks to the development of tissue decellularization technologies, it is nowadays possible to obtain ECM-derived materials from a large variety of tissues. These materials provide a substrate for cell attachment but also a plethora of potential bioinductive signals to the seeded cells, which may influence their differentiation [56,57] all the while showing excellent processability as biological scaffolds. AT is a potentially abundant source of human ECM. Compared to other human adult tissues, white AT is an abundant and available source of human-derived raw-tissue material, which can be obtained from liposuction surgeries. Previous works succeeded in the generation of DAT to provide an inductive microenvironment for stem cells in vitro. Adipogenic differentiation of human adipose-derived stem cells has been achieved previously using this model, however, results were absent for other types of stem cells of non-adipogenic origin [58]. It is convenient to note the importance of the cell donor source in the development of tissue-engineering strategies, as well as the reduction of costs and the need to optimize protein formulation for a large-volume of soft-tissue regeneration for future cell therapies. hDAT is an intact ECM and, according to the literature, shows bioinductive capacity of the components of the natural ECM [58]. Some commercial materials (alginate, gelatin, collagen, fibrin, etc.) are composed of a single ECM protein and are, therefore, biologically insufficient to recreate a microenvironment and 3D cellular organization typical of natural tissues. Besides, while promising for some applications, applying decellularized tissues in their intact form offers limited versatility in terms of tuning scaffold properties, including shape, manageability, stiffness, and degradability, as well as the mode of delivery for in vivo applications [25]. Numerous research groups are applying further processing methods and blending materials to generate customized scaffold formats, such as coatings, foams, hydrogels, and bioinks [26,29,59].

In the present work, we have demonstrated that the substitution of a 25% of the total weight (0.5% (*w*/*v*) of material) in hDAT solid foam offers a viable alternative in terms of physical, mechanical, and biocompatibible properties with stem cell integration and adipogenic differentiation. Thus, the generation as well as mechanical and structural characterization and validation of hDAT solid foams combined with bCOL-I in a proportion 3:1 (respectively) was compatible with the absence of significant differences in the material degradation observed in pure bCOL-I. Moreover, water absorption and storage modulus of hDAT solid foams combined with bCOL-I were maintained with respect to pure hDAT solid foams. Thus, this strategy may save costs and offer an alternative for custom-made solid foam production. From the biological perspective, we sought to use a non-adipogenic stem cell origin to check the adipogenic potential in order to discard the “primed” or committed status of stem cell differentiation tested on previous studies [58]. For this reason, we decided to test the adipogenic potential in primary cultures from human origin using a source of multipotent adult stem cells derived from the neural crest, such as hDPSCs [36,60]. hDPSCs are considered a biological waste with a large potential for cell therapy due to the clear possibility to derive them from autologous tissue [61]. Moreover, hDPSCs are known to be able to differentiate to neural crest progenitors [62], vascular cells [37], and bone cells [39], among other phenotypes. We recently reported that hDPSCs were able to differentiate to osteogenic cells with different efficiencies in decellularized porcine and human AT solid foams, using a solvent extraction method [29]. It remained to be elucidated if hDPSCs were able to differentiate towards AT using hDAT foams generated by a two-step enzymatic digestion method. In the present work, we found that hDAT foams were permissive to allow the survival and proliferation of hDPSCs in a similar way to bCOL-I foams. hDPSCs in adipogenic differentiation media, either in presence of the hDAT 100% pure or 3:1 proportion of hDAT:bCOL-I, were able to increase the mRNA expression of adipocyte-derived lipoprotein lipase (*LPL*) and *C/EBP*, both genes being involved in adipocyte differentiation and intracellular lipid accumulation [47,48]. We also found that PPAR-γ, a nuclear receptor involved in the regulation of several genes associated with growth, differentiation, and regulation of AT [49,63] was more strongly expressed when hDPSCs were cultured in adipogenic media in the presence of hDAT either 100% pure or in the 3:1 proportion of hDAT:bCOL-I solid foams, with respect to bCOL-I ones [64]. To corroborate our results, Oil red staining allowed distinguishing mature adipocytes from other cells types with lipid droplets. We found that a time of merely 14 days of culture of hDPSCs in adipogenic medium in either hDAT or 3hDAT:1bCOL-I foams was sufficient to commit and differentiate hDPSCs into adipocytes. This represents a substantial improvement over traditional adipocyte differentiation methods from hDPSCs in 2D culture, which typically take about four weeks to complete [38].

From the point of view of innovation, our strategy of combining bCOL-I and hDAT ECM materials allows the customization of the hDAT quantity for cell culture in solid foams, and saves production costs by reducing the use of hDAT. We demonstrate that the amount of heterogeneous matrix composition, such as hDAT, can be reduced by using different single protein materials, such as bCOL-I, without significantly altering the potential of enhanced differentiation of hDPSCs towards an adipose phenotype.

## 5. Conclusions

In the present work, we describe a method to generate 3D solid foams fabricated from pure bCOL-I, 100% pure hDAT, or a 3:1 proportion of hDAT:bCOL-I. Moreover, we validate all three solid foams with similar characteristics as 3D substrates compatible with the culture of human stem cells isolated from the dental pulp. We show that at short term the 3D substrates display a remarkably good stem cell integration and survival. Furthermore, at medium term when cultured cells were subjected to adipocyte differentiation during 14 days in hDAT and 3hDAT:1bCOL-I solid foams, *LPL* and *C/EBP* gene expression was increased together with a nuclear localization of PPARγ receptor and Oil red positive staining for lipid droplets. In conclusion our 3D solid foams obtained by hDAT and 3:1 proportion of hDAT:bCOL-I are compatible for human stem cell culture and differentiation towards adipogenic phenotype, with an enhanced efficiency compared to traditional protocols. The present results represent one step forward towards the use of human-derived materials and cells for personalized use in stem cell research and future therapies.

## 6. Patents

This work used the methodology of the patent W02017114902 method for producing a decellularized tissue matrix.

## Figures and Tables

**Figure 1 biology-11-01099-f001:**
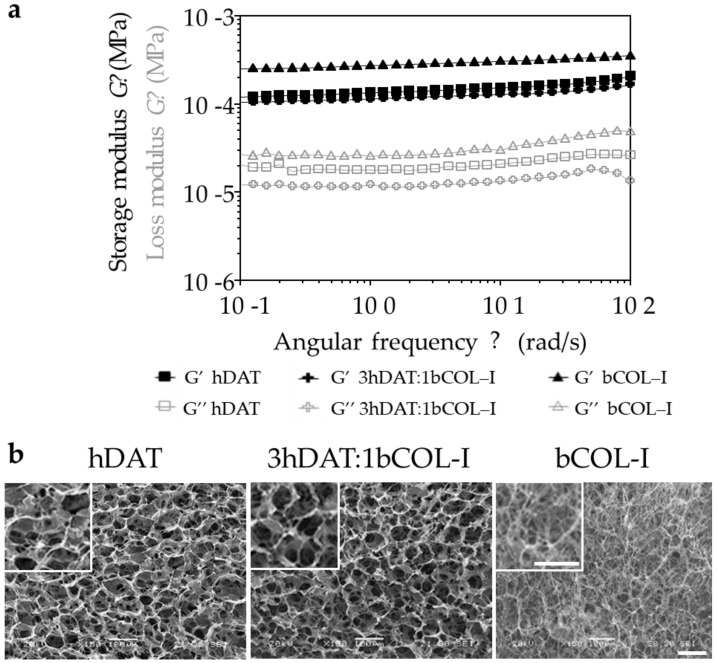
Viscoelastic behaviour and ultrastructural morphology of the solid foams. (**a**) Logarithmic quantification of the viscoelastic behaviour showing storage and loss modulus. (**b**) Ultrastructural images of the different solid foams. Scale bars = 100 μm.

**Figure 2 biology-11-01099-f002:**
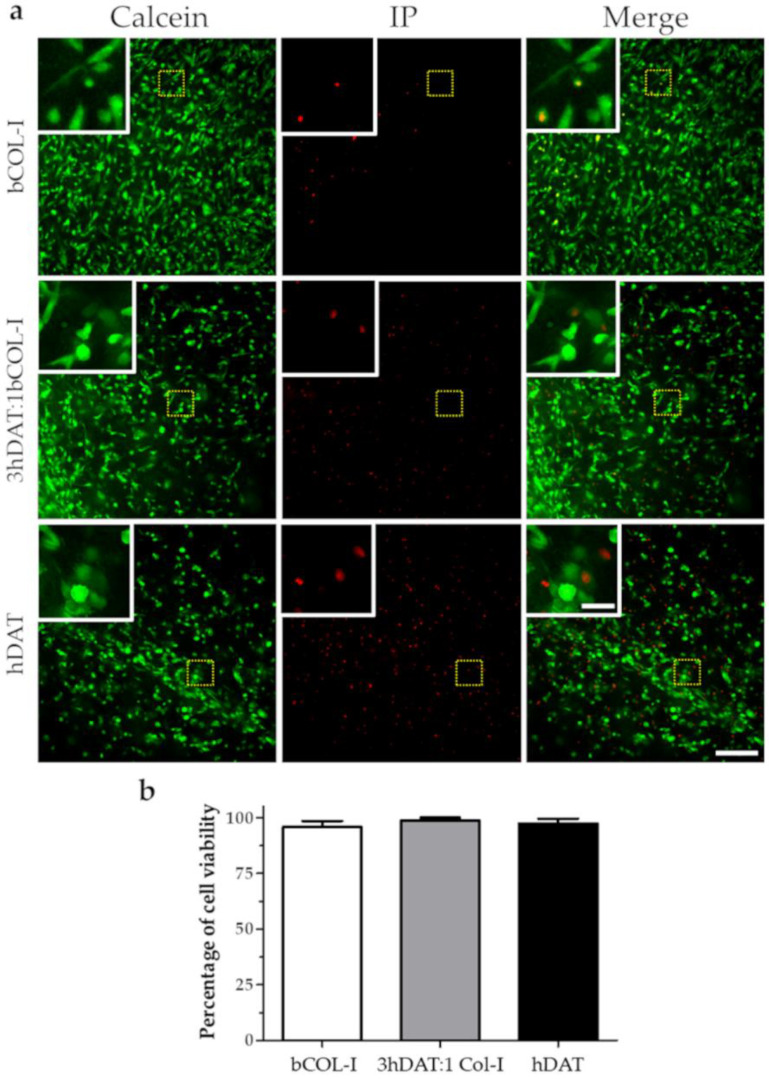
hDPSC viability in solid foams after 72 h. (**a**) Fluorescence microscope images of live (Calcein-AM green) and dead cells (Propidium iodide red). Upper row: bCOL-I, middle row: 3hDAT:1bCOL-I; and lower row: hDAT solid foams (scale bars represent 200 µm, inset 50 µm). (**b**) Quantification of cell viability, data are represented as mean ± SD.

**Figure 3 biology-11-01099-f003:**
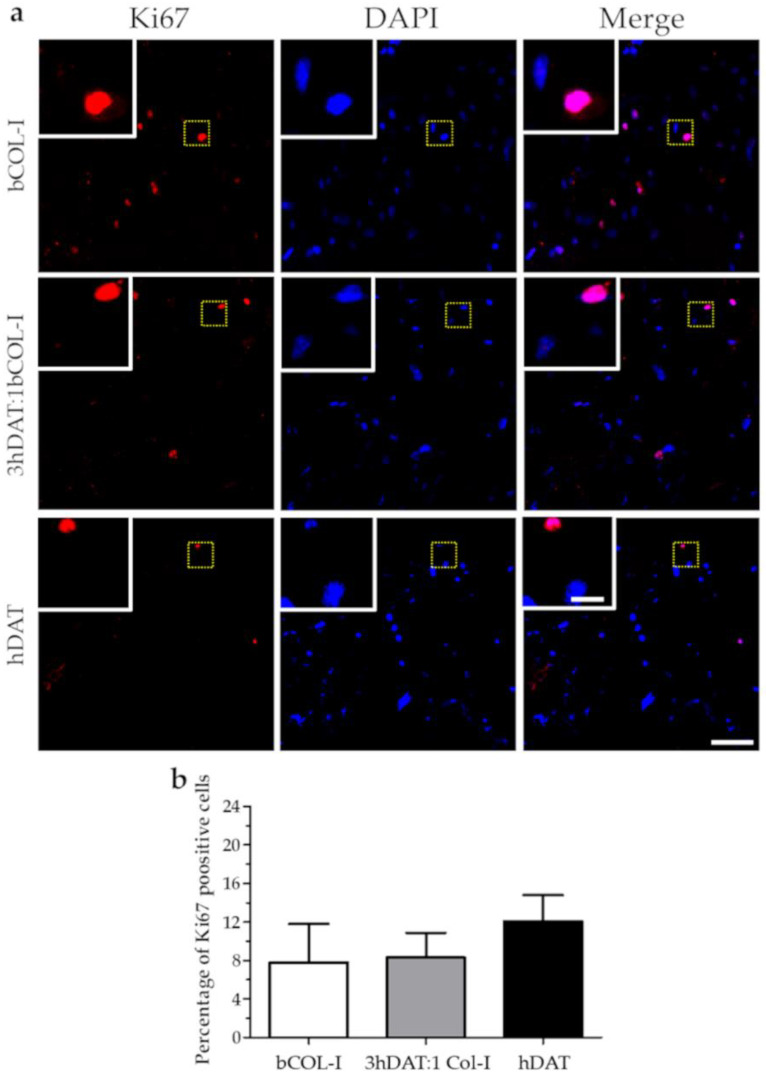
hDPSC proliferation cultured in solid foams after 72 h. (**a**) Fluorescence microscope images of immunofluorescent labeling against Ki-67 (red) and nuclear counterstaining with DAPI (blue). Upper row: bCOL-I, middle row: 3hDAT:1bCOL-I, and lower row: hDAT solid foams (scale bars represent 100 µm, inset 25 µm). (**b**) Quantification of cell viability, data are represented as mean ± SD.

**Figure 4 biology-11-01099-f004:**
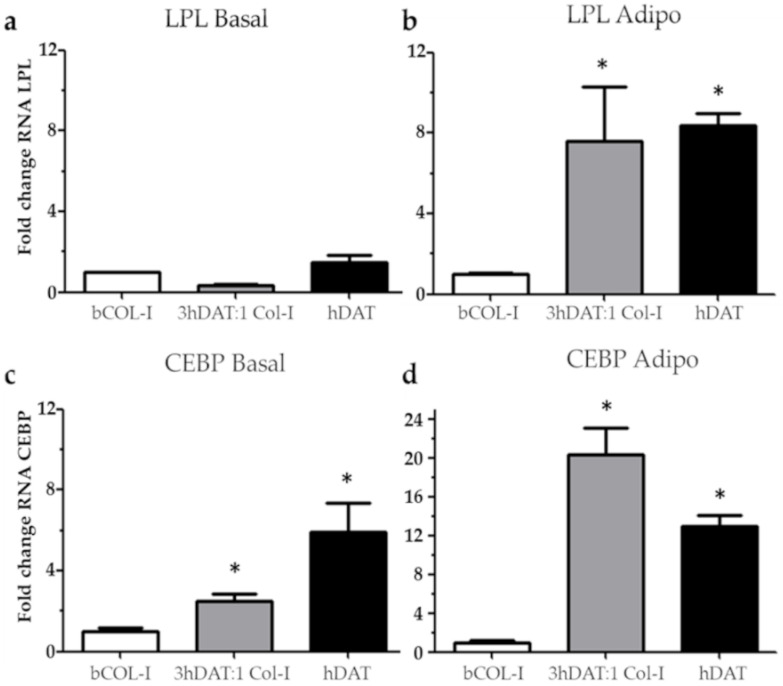
Relative fold change in gene expression by RT-qPCR analysis of hDPSC cultured in bCOL-I, 3 hDAT:1bCOL-I, and hDAT solid foams. (**a**) Fold change of Lipoprotein lipase (LPL) cultured for 14d with basal medium or (**b**) adipogenic medium. (**c**) Fold change of CCAAT/enhancer binding protein (CEBP) cultured for 14 days with basal medium or (**d**) adipogenic medium. Dunn’s pot-hoc test, one-way ANOVA * *p* < 0.05 compared to each respective basal media condition.

**Figure 5 biology-11-01099-f005:**
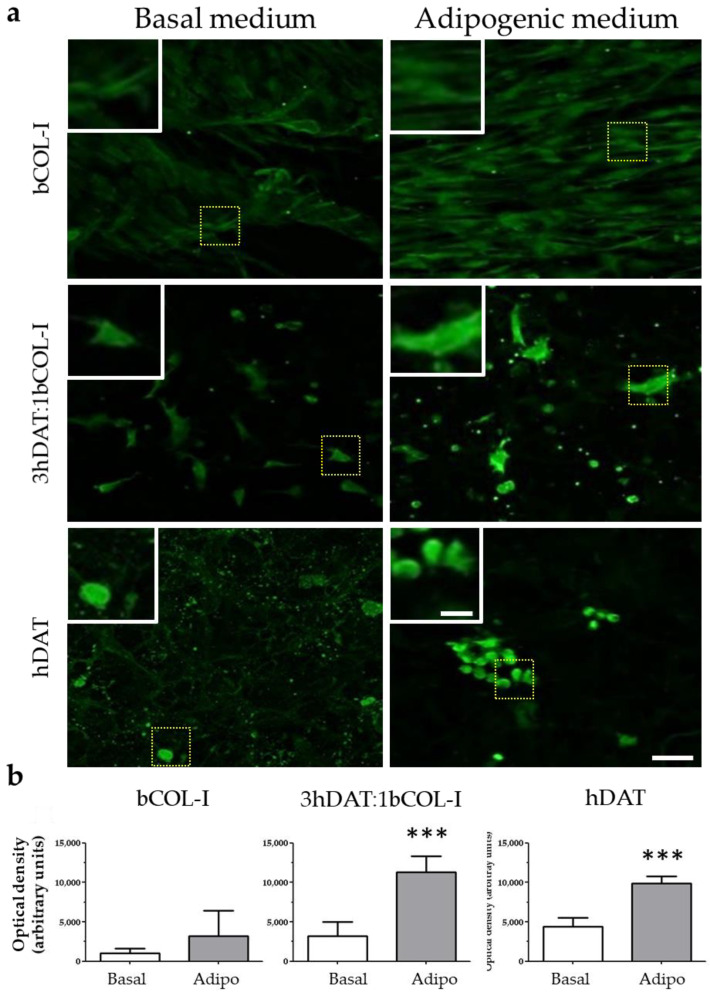
PPAR-γ staining of hDPSCs cultured in solid foams with basal or adipogenic medium for 14 days in culture. (**a**) Images of immunofluorescent labeling against PPAR-γ (green). Upper row: hDAT, middle row: 3hDAT:1bCOL-I, and lower row: bCOL-I solid foams (scale bars represent 100 µm, inset 25 µm). (**b**) Quantification of the PPAR-γ signal of the different conditions. *t*-student *** *p* < 0.001.

**Figure 6 biology-11-01099-f006:**
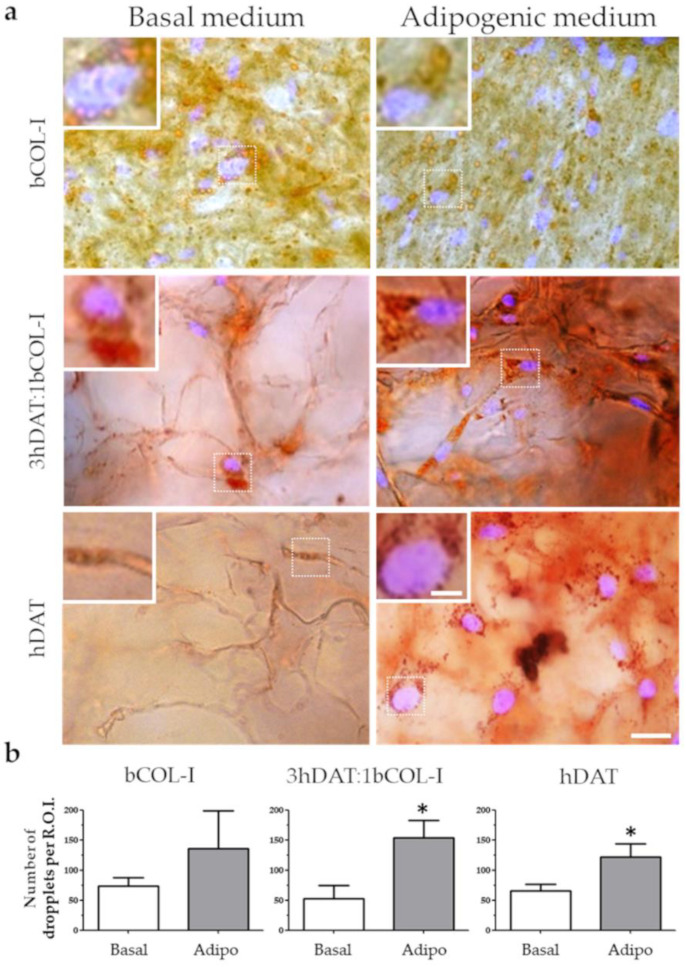
Lipid deposition staining of hDPSCs cultured in solid foams with basal or adipogenic medium for 14 days in culture. (**a**) Images of Oil red (red) counterstained with DAPI. Upper row: bCOL-I, middle row: 3hDAT:1bCOL-I, and lower row: hDAT solid foams (scale bars represent 50 µm, inset 10 µm). (**b**) Quantification of the Oil Red stained droplets of the different culture conditions. Student’s *t*-test * *p* < 0.05.

**Table 1 biology-11-01099-t001:** Properties of hDAT-based solid foams.

ID	Formulation (wt%)	Water Absorption		Average Storage Modulus		Degradation Weight Loss (%)
		(EWC, %)	S Ratio	G′ (Pa)	G″ (Pa)	
hDAT	100% hDAT	92.4 ± 0.9	12.2 ± 1.6	152.8 ± 15.6	21.1 ± 3.2	22.9 ± 11.9
3hDAT:1 bCOL-I	75% hDAT: 25% bCOL-I	92.1 ± 0.7	11.7 ± 1.1	128.0 ± 11.1	13.6 ± 2.1	26.10 ± 11.5
bCOL-I	100% bCOL-I	90.3 ± 0.1	9.3 ± 0.1	302.9 ± 22.2	32.7 ± 7.0	61.95 ± 15.99

**Table 2 biology-11-01099-t002:** List of primers used in this study.

Primers	Sequence 5′-3′	Annealing	Amplicon (bp)
*β-ACTIN* (upstream)	GTTGTCGACGACGAGCG	58.5	93
*β-ACTIN* (downstream)	GCACAGAGCCTCGCCTT	59.7	93
*GAPDH* (upstream)	CTTTTGCGTCGCCAG	60.3	131
*GAPDH* (downstream)	TTGATGGCAACAATATCCAC	60.8	131
*CEBP* (upstream)	AGCCTTGTTTGTACTGTATG	54.3	199
*CEBP* (downstream)	AAAATGGTGGTTTAGCAGAG	58.3	199
*LPL* (upstream)	ACACAGAGGTAGATATTGGAG	53.8	143
*LPL* (downstream)	CTTTTTCTGAGTCTCTCCTG	52.9	143

## Data Availability

All data are available upon request.

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
