# Peer review of "Enhanced Adipogenic Differentiation of Human Dental Pulp Stem Cells in Enzymatically Decellularized Adipose Tissue Solid Foams"

_biology, 2022, doi:10.3390/biology11081099_

Round 1

Reviewer 1 Report

The present paper is well written and designed. I suggest to the authors to add a graphical abstract to resume the entire paper and simplify the readers.

Then i also suggest to perfom some experiments to evaluate the adipogenic markers (for e.g. PPARg or FABP4...) by means WB or RT-PCR or other methodologies. Authors shiuld add in the introduction section some informations about dental pulp stem cells and their important role in the regenerative medicine (for e.g. doi: 10.3390/biomedicines10020403;  doi: 10.3390/ma13010130). 

Author Response

Reviewer 1:

Comments and Suggestions for Authors

The present paper is well written and designed. I suggest to the authors to add a graphical abstract to resume the entire paper and simplify the readers.

ANSWER: We thanks to the author the comment. We prepared the following graphical abstract and express our apologies for its loss during the uploading procedure:

Then I also suggest to perfom some experiments to evaluate the adipogenic markers (for e.g. PPARg or FABP4...) by means WB or RT-PCR or other methodologies.

ANSWER: Thank you for the improvement suggestion. We need to prepare more SFs and unfortunately we had a stock rupture of one required compound. We are expecting to receive it and continue the SFs production, sterilization and cell seeding and differentiation. We are planning to perform FABP4 staining to complement the differentiation phenotype comparing basal and adipogenic culture conditions in order to provide additional qualitative and quantitative data of cell differentiation. We will communicate the results to the reviewer as soon as possible and we sincerely apologize for this unexpected and unavoidable delay.

Authors shiuld add in the introduction section some informations about dental pulp stem cells and their important role in the regenerative medicine (for e.g. doi: 10.3390/biomedicines10020403;  doi: 10.3390/ma13010130). 

ANSWER: We thank again the reviewer for this valuable suggestion. We added the recommended bibliography (we marked it in red in the text, lines 134-135) to underscore the relevance of dental pulp stem cells in regenerative medicine.

Reviewer 2 Report

1. The author's previous research (Eur Cell Mater 2022, 43, 112–129)  has confirmed that hDAT can promote adipogenic differentiation of HDPs. This research is only mixed with COL-I, so the novelty is not enough.

2.Why choose the ratio of hDAT and COL-I 3:1? What about other ratios?

3. Are there significant differences in PPAR-É£ staining and Lipid deposition staining results between three groups of hDAT, 3hDAT:1bCOL-I and bCOL-I ?

4. There are also some format errors, please proof-read carefully. For example, line 451 "table 2" should be "Figure 1", lines 509,512,514 figure3 a, b, c should be "figure4 a, b, c" .

Author Response

Answer to reviewer 2:

1.-The author's previous research (Eur Cell Mater 2022, 43, 112–129) has confirmed that hDAT can promote adipogenic differentiation of HDPs. This research is only mixed with COL-I, so the novelty is not enough.

ANSWER: We thank the reviewer for the pertinent comment as we think that it might be shared by other potential readers. It is true that adipogenic differentiation on hDAT had been already reported before, but this is not the same hDAT material that was employed in the Eur. Cell Mater. 2022 paper. Various methods can be used for decellularization of almost all types of tissues and each involves a combination of different physical, ionic, chemical, or enzymatic treatments. Moreover, it is widely accepted that each treatment affects the biochemical composition, ultrastructure, and mechanical behavior of the remaining extracellular matrix in a unique way. 

Effectively, in the recent past our group has developed four decellularized adipose tissue (DAT) materials obtained from two different tissue sources (human and porcine) and decellularized by two different protocols (enzymatic and organic solvent)(Cicuendez et al, 2021). In our previous recent research (Eur Cell Mater 2022, 43, 112–129) the human DAT was obtained by organic solvent decellularization, so the material was different in both protein composition and residual lipids (Cicuendez et al, 2021).

The novelty of the present manuscript is the fabrication and decellularisation of the human material, which is based on the two-step enzymatic method that we specifically designed as the most effective and simple method of decellularisation for human adipose tissue, avoiding the use of any organic solvent (Madarieta et al., Patent WO/2017/114902, 2017, Cicuendez et al 2021). Additionally, this is the first time that we describe the processability of this specific material as a solid foam, avoiding organic solvents and successfully obtaining composite materials to analyze the effect of hDAT dosage on stem cell differentiation in 3D cultures. We tried to clarify the message in the text of the manuscript.

2.Why choose the ratio of hDAT and COL-I 3:1? What about other ratios?

ANSWER: Biologic scaffold materials might provide a substrate for cell attachment and a source of bioinductive signals which may influence cell behavior in culture similarly to what happens in native tissues. Collagen type-I is one of the most explored and commercialized extracellular matrix materials and the principal structural protein of the extracellular matrix in most human tissues. As a single protein and structural material, in our experience, collagen type-I is an ideal control or reference to investigate more complex and bioinductive materials such as decellularized intact extracellular matrices.

In our work, firstly, we showed the excellent processability of human decellularized adipose tissue (hDAT) as solid foams (5% w/v) that met the structural requirements (swelled properly without losing their geometry). However, related to the bioinductive effect of hDAT, we aimed at investigating the rate of hDATthat may induce stem cell differentiation and we substituted hDAT with collagen type-I in a manner that did not incorporate important structural changes on the resultant solid foams. This strategy allowed us to observe the composition effects of the hDAT on the seeded stem cells. During the rheological studies we observed that the substitution of a 25% of hDAT by collagen type-I was the maximum amount that we could introduce in the formulation of the solid foams without giving rise to structural changes (see figure below).  In summary, the reason behind the ratio 3:1 of hDAT and collagen type-I was to replace hDAT, all the while avoiding the introduction of structural changes in the solid foams, to focus the investigation mainly on the effective hDAT/collagen of hDAT that may still induce stem cell adipogenic differentiation.

Figure 1.- Whole rheological study results of hDAT and collagen type-I solid foams obtained by several ratios. Note the important structural contribution of collagen type-I in ratios greater than 25%.

Anyhow, we totally agree with the reviewer that it will be interesting to investigate stem cell behavior with other ratios of hDAT:Collagen type-I. We incorporated here the viability assays of Calcein and caspase-3 viability assays, where we did not observe any significant differences, as shown below:

Figure 2.

Table Analyzed

All SFs Calcein

Kruskal-Wallis test

P value

0.0712

Exact or approximate P value?

Gaussian Approximation

P value summary

ns

Do the medians vary signif. (P < 0.05)

No

Number of groups

5

Kruskal-Wallis statistic

8.624

Calcein-AM and Propidium iodide staining in hDPSC to assess cell viability cultured in solid foams for 72h (A-F). (A-E) Fluorescence microscope images of live (Calcein, green) and dead cells (Propidium iodide, red): (A) hDAT, (B) 3hDAT:1bCOL-I, (C) 2hDAT:2bCOL-I, (D) 1hDAT:3bCOL-I and (E) bCOL-I solid foams (scale bars represent 100 µm). (F) Viability (%) obtained by Image-Pro analysis. Kruskal-Wallis with Dunn’s Multiple Comparison Test.

Table Analyzed

All SFs Ki67

Kruskal-Wallis test

P value

0.5505

Exact or approximate P value?

Gaussian Approximation

P value summary

ns

Do the medians vary signif. (P < 0.05)

No

Number of groups

5

Kruskal-Wallis statistic

3.044

Proliferation of hDPSC cultured in solid foams for 72h (A-F). (A-E) Fluorescence microscope images of primary antibody Ki-67 (green) and nucleus with DAPI (blue) stained cells. (A) hDAT, (B) 3hDAT:1bCOL-I, (C) 2hDAT:2bCOL-I, (D) 1hDAT:3bCOL-I and (E) bCOL-I solid foams (scale bars represent 100 µm). (F) Proliferation (%) obtained by Image-Pro analysis. Kruskal-Wallis with Dunn’s Multiple Comparison Test.

  1. Are there significant differences in PPAR-É£ staining and Lipid deposition staining results between three groups of hDAT, 3hDAT:1bCOL-I and bCOL-I ?

ANSWER: Thank you for the question. Here we provide a table summarizing the different combinations of PPAR-g analysis respect to control bCOL-I (either in basal or adipogenic conditions) and lipid deposition staining:

Table Analyzed

PPAR-É£

One-way analysis of variance

P value

0.0002

P value summary

***

Are means signif. different? (P < 0.05)

Yes

Number of groups

6

F

5.385

R squared

0.2173

Bartlett's test for equal variances

Bartlett's statistic (corrected)

54.74

P value

< 0.0001

P value summary

***

Do the variances differ signif. (P < 0.05)

Yes

ANOVA Table

SS

df

MS

Treatment (between columns)

1480000000

5

296000000

Residual (within columns)

5332000000

97

54970000

Total

6812000000

102

Dunnett's Multiple Comparison Test

Mean Diff.

q

Significant? P < 0.05?

Summary

95% CI of diff

bCOL-Basal vs 3hDAT_bCOL-Basal

-2188

0.8441

No

ns

-8811 to 4435

bCOL-Basal vs hDAT-Basal

-3424

1.365

No

ns

-9830 to 2983

hDAT-Basal vs 3hDAT_bCOL-Basal

1236

0.4706

No

ns

-5475 to 7947

3hDAT_bCOL-Basal vs 3hDAT_bCOL-Adipo

-8087

3.079

Yes

*

-14800 to -1376

bCOL-Adipo vs 3hDAT_bCOL-Adipo

-8051

3.166

Yes

**

-14550 to -1554

bCOL-Adipo vs hDAT-Adipo

-6637

2.681

Yes

*

-12960 to -312.4

hDAT-Adipo vs 3hDAT_bCOL-Adipo

-1415

0.5715

No

ns

-7739 to 4910

As can be observed above, there are statistical significant differences (p=0.0002). Moreover, formulations containing hDAT (hDAT or 3hDAT:bCOL-I) in adipogenic medium increase PPAR-g staining.

Table Analyzed

Oil-Red

One-way analysis of variance

P value

0.0164

P value summary

*

Are means signif. different? (P < 0.05)

Yes

Number of groups

6

F

3.524

R squared

0.4338

ANOVA Table

SS

df

MS

Treatment (between columns)

40440

5

8088

Residual (within columns)

52790

23

2295

Total

93230

28

Dunnett's Multiple Comparison Test

Mean Diff.

q

Significant? P < 0.05?

Summary

95% CI of diff

bCOL-Basal vs 3hDAT_bCOL-Basal

20.87

0.5964

No

ns

-73.78 to 115.5

bCOL-Basal vs hDAT-Basal

7.200

0.2482

No

ns

-71.28 to 85.68

hDAT-Basal vs 3hDAT_bCOL-Basal

13.67

0.4034

No

ns

-77.98 to 105.3

3hDAT_bCOL-Basal vs 3hDAT_bCOL-Adipo

-101.5

2.900

Yes

*

-196.1 to -6.818

bCOL-Adipo vs 3hDAT_bCOL-Adipo

-18.00

0.5941

No

ns

-99.97 to 63.97

bCOL-Adipo vs hDAT-Adipo

13.80

0.4555

No

ns

-68.17 to 95.77

hDAT-Adipo vs 3hDAT_bCOL-Adipo

-31.80

1.050

No

ns

-113.8 to 50.17

As can be observed above, there are statistically significant differences (p=0.0164). The presence of hDAT in the formulation rendered hDPSCs more responsive to adipogenic induction media, contrary to b-COL.

  1. There are also some format errors, please proof-read carefully. For example, line 451 "table 2" should be "Figure 1", lines 509,512,514 figure3 a, b, c should be "figure4 a, b, c" .

ANSWER: We apologize for the errors. We have double-checked the manuscript thoroughly to correct any errors. The written sentence of lines 396-399: “Solid foams showed higher average storage modulus with the increase in bCOL-I quantity in the formulation: 152.8±15.6 Pa, 128.0±11.1 Pa and 302.9±22.2 Pa, for hDAT, 3hDAT:1 bCOL-I and bCOL-I, respectively (Table 2).” We refers to the data of Table 2. We placed it immediately after Figure 1 to clarify the message and present all the data together to facilitate the comprehension to the potential readers.

Round 2

Reviewer 1 Report

All my comments have been adressed.

Reviewer 2 Report

Accept in present form